# Using Peoples' Perceptions to Improve Conservation Programs: The Yellow-Shouldered Amazon in Venezuela

**Ada Sánchez-Mercado [1,2,\*]**, **Oriana Blanco [1], Bibiana Sucre-Smith [1]**,
**José Manuel Briceño-Linares [1], Carlos Peláez [1] and Jon Paul Rodríguez [1,3,4]**

[1] Provita, Calle La Joya, Edificio Unidad Técnica del Este, Piso 10, Oficina 30, Chacao, Caracas 1060, Venezuela;
oblanco@provitaonline.org (O.B.); bsucre@provitaonline.org (B.S.-S.); jbriceno@provitaonline.org (J.M.B.-L.);
capelaez@gmail.com (C.P.); JonPaul.RODRIGUEZ@ssc.iucn.org (J.P.R.)

[2] School of Biological, Earth and Environmental Sciences, University of New South Wales, Kensington,
NSW 2052, Australia

[3] Centro de Ecología, Instituto Venezolano de Investigaciones Científicas, Apdo. 20632,
Caracas 1020-A, Venezuela

[4] IUCN Species Survival Commission, 28 rue Mauverney, CH-1196 Gland, Switzerland

\* Correspondence: ay.sanchez.mercado@gmail.com; Tel.: +61-48-1203171

**Abstract:** The perceptions and attitudes of local communities help understand the social drivers of unsustainable wildlife use and the social acceptability of conservation programs. We evaluated the social context influencing illegal harvesting of the threatened yellow-shouldered Amazon (*Amazona barbadensis*) and the effectiveness of a longstanding conservation program in the Macanao Peninsula, Margarita Island, Venezuela. We interviewed 496 people from three communities and documented their perceptions about (1) status and the impact of threats to parrot populations, (2) acceptability of the conservation program, and (3) social processes influencing unsustainable parrot use. Approval of the program was high, but it failed to engage communities despite their high conservation awareness and positive attitudes towards the species. People identified unsustainable use as the main threat to parrots, but negative perceptions were limited to selling, not harvesting or keeping. Harvesters with different motivations (keepers, sellers) may occur in Macanao, and social acceptability of both actors may differ. Future efforts will require a stakeholder engagement strategy to manage conflicts and incentives to participation. A better understanding of different categories of harvesters, as well as their motives and role in the illegal trade network would provide insights to the design of a behavior change campaign.

**Keywords:** conservation management; conservation threats; drivers of extinction; illegal wildlife trade; parrot conservation; Psittacidae conservation; threatened species; unsustainable use of wildlife

## 1. Introduction

Conservation programs often focus on reducing the unsustainable use of wildlife in highly complex social–ecological environments, where local communities are key actors in both the trade chain and the conservation actions implemented [1–4]. Studying the perceptions and attitudes of local communities towards the unsustainable use of wildlife has been key to understanding social and cultural drivers of unsustainable use [5,6], social acceptability of conservation management [7–9], and the design of culturally suitable and more tenable conservation actions [10].

Due to the cultural nature of the illegal parrot trade, local people's perceptions are particularly important to assess performance of conservation programs aimed at tackling this threat. People have

been keeping parrots as pets for centuries [11] (Psittaciformes, which include parrots, macaws, parakeets, parrotlets, and cockatoos), and today 28% (111 of 398) of extant species are listed as threatened on the International Union for Conservation of Nature's Red List of Threatened Species [12]. Unsustainable use, including harvest, trade, and keeping, is highly influenced by the species' attractiveness to humans [13,14], but also by cultural and social factors: parrots owners often regard their animals as "family members", perceived and treated as children [15]. This social role may also influence understanding of psittacid conservation challenges and attitudes towards conservation actions.

Here, we evaluate the social context influencing the use of the yellow-shouldered Amazon (*Amazona barbadensis*) and the effectiveness of a longstanding conservation program led by Provita and aimed at restoring their population in the Macanao Peninsula, Margarita Island, Venezuela. The yellow-shouldered Amazon ("parrots" hereafter) is classified as vulnerable internationally [16] and endangered regionally [17], due to the capture of nestlings for the pet trade (both domestic and international) [18] and the destruction of nesting and feeding habitats [19]. The main population (ca. 1600 individuals) inhabits Macanao Peninsula [20]. Provita is a Venezuelan non-governmental organization that has implemented the Yellow-Shouldered Amazon Conservation Program in Macanao over the last 31 years. The program includes school-age environmental education activities, and full-time surveillance of natural and artificial nests in the main breeding site of this parrot population (La Chica). The Ecoguardians, a cooperative of local young people recruited, trained, and hired by Provita, have implement most of these actions in the field [20,21]. However, after 31 years of implementation, illegal harvesting persists, and it is unclear whether this unsustainable use points towards the need to strengthen enforcement strategies [20] or aim for a more holistic approach focused on behavioral change. We specifically explore local perceptions about (1) the status and impact of threats to parrot populations, (2) acceptability of the conservation program in terms of support and responsibilities, and (3) social processes influencing unsustainable parrot use and the performance of conservation actions.

## 2. Materials and Methods

### 2.1. Study Area and Socioeconomic Context

Macanao Peninsula is located in the western portion of Margarita Island and is less developed for tourism than the eastern part, resulting in ecosystems that are in relatively good condition (Figure 1) [20]. By 2011, there were approximately 24,419 inhabitants in Macanao (a tenth of Margarita's population). Employment opportunities are scarce, with fishing being the primary economic activity [22].

### 2.2. Interview Instruments and Survey Process

Between March and September 2017, we interviewed 496 people from three communities across Macanao (Boca del Río, El Horcón, and Robedal; Figure 1b) using a self-reporting questionnaire. All participants were adults (>18 years old), and had lived in Macanao for at least one year. We obtained verbal informed consent from each subject, after explaining the research objectives and assuring subjects that information would be used only for research, and presented the data in aggregate analyses, protecting each participant's identity [23]. The survey protocol was approved by the Laboratory of Political Ecology of the Venezuelan Institute of Scientific Research (February 2017), who acted as the external ethical committee. Households were chosen randomly from community maps, by selecting every fourth house. The sample size represented 21–30% of households in each community.

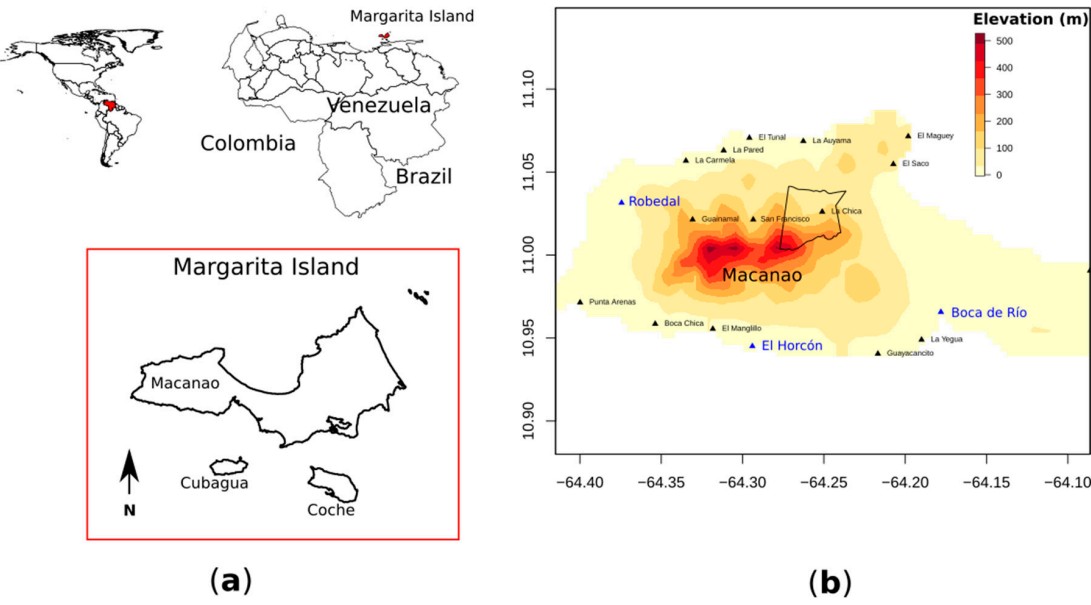

**Figure 1.** Study area. (**a**) Relative position of Venezuela, Margarita Island, and the Macanao Peninsula. (**b**) Elevation gradient in Macanao. Communities surveyed are highlighted in blue. The main nesting site where Provita implements nest monitoring and surveillance, Hato San Francisco, Quebrada La Chica, is delimited by a black polygon.

We evaluated the general socioeconomic context of participants by asking about their age, gender, level of education, employment status, source of income, and whether this income was enough to cover family monthly expenses. The survey instrument evaluated three distinct aspects related to conservation practice [1]: (1) ecological outcomes, (2) acceptability of conservation management, and (3) social processes influencing the effectiveness of conservation actions (Table 1). To assess perceived ecological outcomes, we evaluated three aspects: awareness of conservation status, perceived threats and their impact on wild populations, and the success of surveillance in preventing fledgling poaching (Table 1). We measured awareness by asking two closed questions: whether the participants keep parrots at home (owners) or not (non-owners), and if they think there are more parrots in captivity than in the wild (yes/no). We evaluated the perceived impact on the wild population by asking two closed questions: "Do you think that the wild parrot population will go extinct in the next 10 years?" and "Do you think that the wild parrot population is stable, declining, or increasing?" To assess people's knowledge about threats faced by the wild parrot population, we asked an open question—"What is the main threat faced by parrots?"—and then reclassified the answers into four categories: "unsustainable use", "deforestation", "drought", and "predators." We asked "Where do you think your parrot comes from?" as a closed question, with the names of the most important nesting sites as options. We used this question as a measure of surveillance effectiveness, as La Chica has been the only nesting site under protection during the last 31 years (Table 1).

To measure the acceptability of conservation management, we evaluated three aspects: support for the conservation program, perceptions of other stakeholders in the process, and perceived responsibilities and roles. We measured support for the conservation program by asking awareness of Provita's work with a closed question "Do you know Provita's work?", and whether it addresses the main threats to the species: "What do you think is the main conservation problem that Provita cares for?" For this latter question, we reclassified the answers into the same four categories we used to assess people's knowledge about threats, so that responses were comparable. Given that the Ecoguardians are a key stakeholder, we inquired about perceptions towards Ecoguardians with an open question, and then reclassified the answers into positive and negative perceptions. To assess perceived responsibilities, we asked which are the institutions responsible for parrot conservation (Provita, communities, or government authorities). To evaluate the role that people have in the illegal

parrot trade chain, we used an open question "How did you get your parrot?" We then reclassified the answers into four categories "harvested", "bought", "rescued", and "present/gift" (Table 1).

**Table 1.** Evaluation of the yellow-shouldered Amazon Conservation Program, based on perceptions in three communities of the Macanao Peninsula, Margarita Island, Venezuela.

| Conservation Issue | Aspect Evaluated | Questions |
|---|---|---|
| Ecological outcomes of conservation | Awareness about species conservation status | Do you keep a parrot at home? |
| | | Do you think that there are more parrots in captivity than in the wild? |
| | Perceived impact on wild population | Do you think that the wild parrot population will go extinct in the next 10 years? |
| | | Do you think that the wild parrot population is stable, declining, or increasing? |
| | Perception of species threats Effects of the conservation action (surveillance) | What is the main threat faced by parrots? What is the main location for fledgling extraction? |
| Acceptability of conservation management | Support for conservation program | Do you know Provita's work? |
| | | What do you think is the main conservation problem that Provita cares for? |
| | Perceptions of other stakeholders in the process | What do you think about the work of Ecoguardians? |
| | Perceived responsibilities and roles | Who is the entity/organization responsible for parrot conservation? How did you get your parrot? (role in the trade chain) |
| Social processes affecting conservation actions | Social value of the species | What does your parrot mean to you? Who gave you your parrot? |
| | Attitudes towards stages in the trade chain | Do you agree with this statement? "I will always want to keep a parrot as pet." Fledgling parrot extraction is … Do you report poachers? Selling fledgling parrots is … |

To understand social processes affecting conservation action, we evaluated two aspects: the social value of parrots and attitudes towards harvesting, selling, and keeping. We used an open question "What does your parrot mean to you?", and then we reclassified the answers into three categories: "pet", "a family member", or "symbol." We also asked whether their parrot was provided by a member of the community, a relative, or an outsider. We used a statement to measure attitudes toward keeping parrots as pets, which was "I will always want to keep a parrot as pet"; we assessed answers on a five-point scale that ranged from 1 (strongly disagree) to 5 (strongly agree). We asked about attitudes towards reporting poaching with a closed question "Do you report poachers?", and in the instances with negative replies we additionally asked "Why not?" and aggregated the answers into four categories: "denounce", "not denounce", "indifferent", and "support." We evaluated attitudes toward extraction and selling using open statements, such as "Fledgling parrot extraction is … " and "Selling fledgling parrots is … ", and then we classified the answers into positive or negative attitudes. (Table 1). Finally, we asked how much their parrots were worth in national currency, and converted it into USD using the weekly mean of the currency exchange rate, and how many individuals they currently keep captive.

We summarized the responses (number of records and percentages) for each variable at the community level and for the overall sample.

## 3. Results

### 3.1. Characteristics of the Sample

Survey participants were 69% female, with a mean age of 43.7 years old (SD = 15.0). Half of survey participants were unemployed (53%), and 51% of them had a high school diploma, while 26% have had university studies (Table 2).

Family income comes mainly from government social support (34%), salary (24%), or retirement pension (23%). For the majority of participants (79%), their income was not enough to cover monthly basic expenses (Table 2). Among the communities surveyed, El Horcón had the most critical socioeconomic condition, with the highest unemployment rate, more dependence on government help (70%), and a lower education level (only 8% of participants hold a university degree, compared to 35% in Boca del Río).

### 3.2. Perceptions about Species Conservation Status and Ecological Outcomes

Twenty-two percent of participants keep at least one parrot at home (Figure 2a). The majority (79%) believed that there are more parrots in the wild than in captivity. Although most participants (80%) believed that wild populations are decreasing, 53% thought that it may lead to extinction in 10 years (Figure 2a).

Unsustainable use was identified as the main threat to parrots (69% of participants; Figure 2b), and people believed that harvest occurred mainly at sites other than La Chica (61% of participants; Figure 2b).

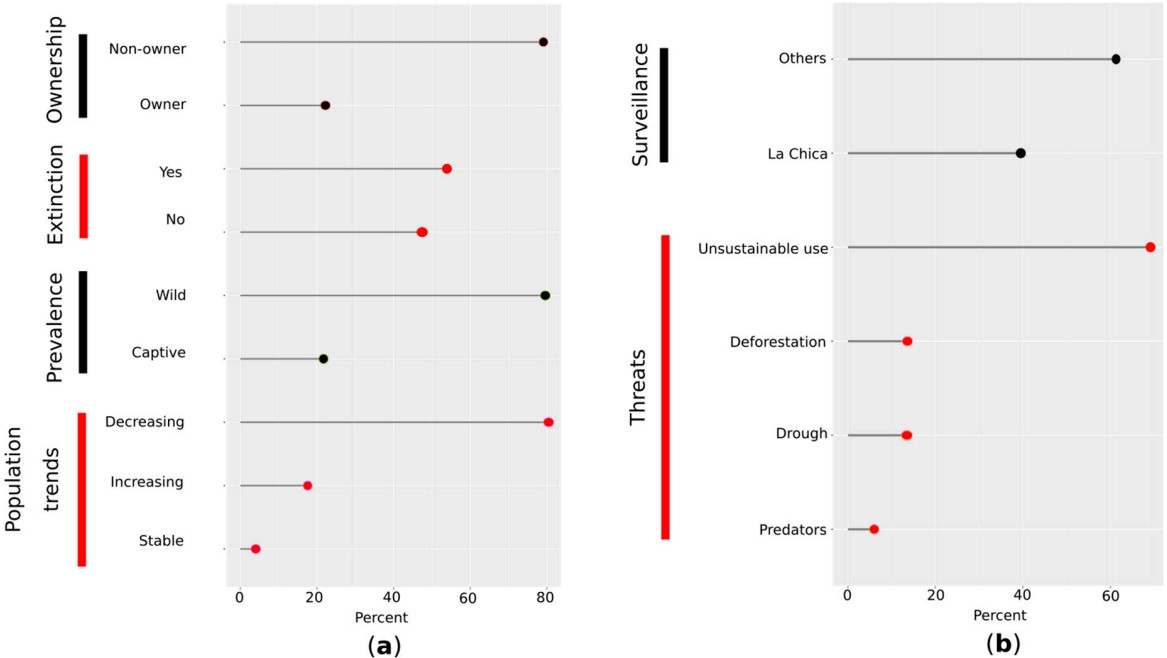

**Figure 2.** Perceived conservation outcomes from Macanao inhabitants regarding (**a**) population status of the yellow-shouldered Amazon and (**b**) main threats and the impact of nest surveillance.

**Table 2.** Socio-economic characteristics of the three communities surveyed in Macanao Peninsula, Margarita Island, Venezuela. Percentage of people by gender, employment status, educational level, income level, and source of income are shown for each community (number of answers in each category). Also, mean and standard deviation of age by community is shown.

| Community | Gender | | Employment Status | | Age | | Education Level | | | | Income Level | | | | | Source of Income | | | |
|---|---|---|---|---|---|---|---|---|---|---|---|---|---|---|---|---|---|---|---|
| | F | M | Unemployed | Employed | Mean | SD | None | Primary | High School | University | All | Almost All | The Half | Few | Nothing | Social Program | Own Revenues | Retirement Pension | Salary |
| Boca del Río | 64 (190) | 36 (107) | 48 (141) | 52 (151) | 47.7 | 14.6 | 1 (3) | 14 (42) | 50 (147) | 35 (102) | 0 (0) | 2 (5) | 8 (23) | 28 (84) | 62 (184) | 26 (66) | 19 (48) | 26 (65) | 29 (73) |
| El Horcón | 84 (41) | 16 (8) | 68 (32) | 32 (15) | 43.4 | 15.8 | 2 (1) | 31 (15) | 59 (29) | 8 (4) | 2 (1) | 16 (8) | 20 (10) | 37 (18) | 24 (12) | 70 (33) | 2 (1) | 19 (9) | 9 (4) |
| Robledal | 74 (111) | 26 (39) | 58 (84) | 42 (62) | 44.0 | 15.1 | 5 (7) | 31 (45) | 50 (73) | 15 (22) | 5 (8) | 16 (23) | 16 (24) | 31 (45) | 32 (46) | 36 (40) | 27 (79) | 18 (94) | 19 (98) |
| All | 69 (342) | 31 (154) | 53 | 47 | 43.7 | 15.0 | 2 | 21 | 51 | 26 | 2 | 7 | 12 | 30 | 49 | 34 | 19 | 23 | 24 |

*3.3. Acceptability of Conservation Management*

　　Perceptions about Ecoguardians were mixed, with 47% of participants holding negative opinions, mostly because they believed Ecoguardians participate in poaching (Figure 3).

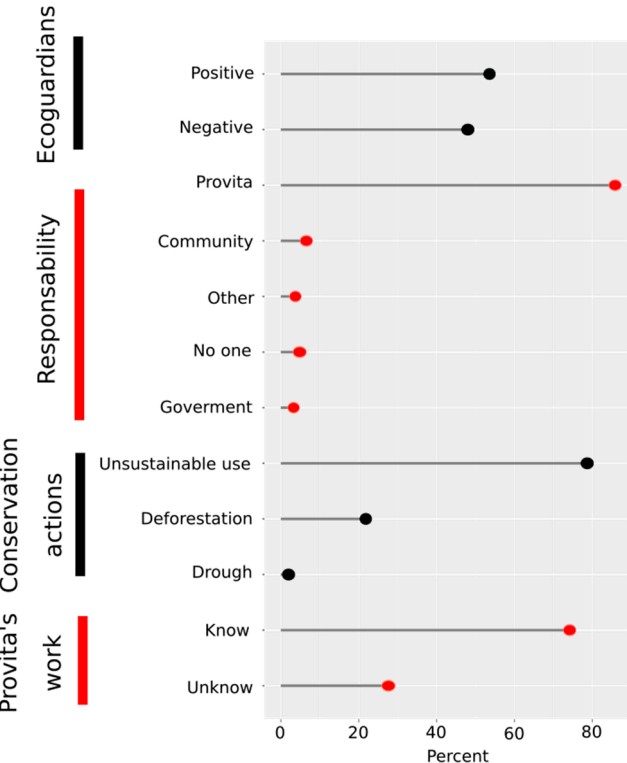

**Figure 3.** Acceptability of conservation management in three communities of the Macanao Peninsula regarding the yellow-shouldered Amazon Conservation Program.

　　People had a high level of awareness of Provita's work (73% of participants) and perceived that their conservation actions are coherent with the primary threat, being focused on the reduction of unsustainable harvest (68%; Figure 3). For the majority of participants, Provita is the organization responsible for parrot conservation (85%), while the role or responsibility of the government is low (9%) and the community even lower (6%; Figure 3).

*3.4. Social Processes*

　　Affective values—parrots as part of the family—were predominant (82%), and in most cases, parrots were presents from relatives (56%) or provided by people from the community (40%). Most interviewees participated in the trade chain as keepers, whether accepting parrots as presents (39%), directly buying (26%), or rescuing abandoned fledglings (21%). About 15% of interviewees admitted being directly involved in extracting their parrot from the wild (Figure 4a). In a few instances (4%), captive parrots came from people outside of Macanao (Figure 4a). Half of interviewees would not report poachers, because either they are relatives or they feared retaliation (Figure 4a).

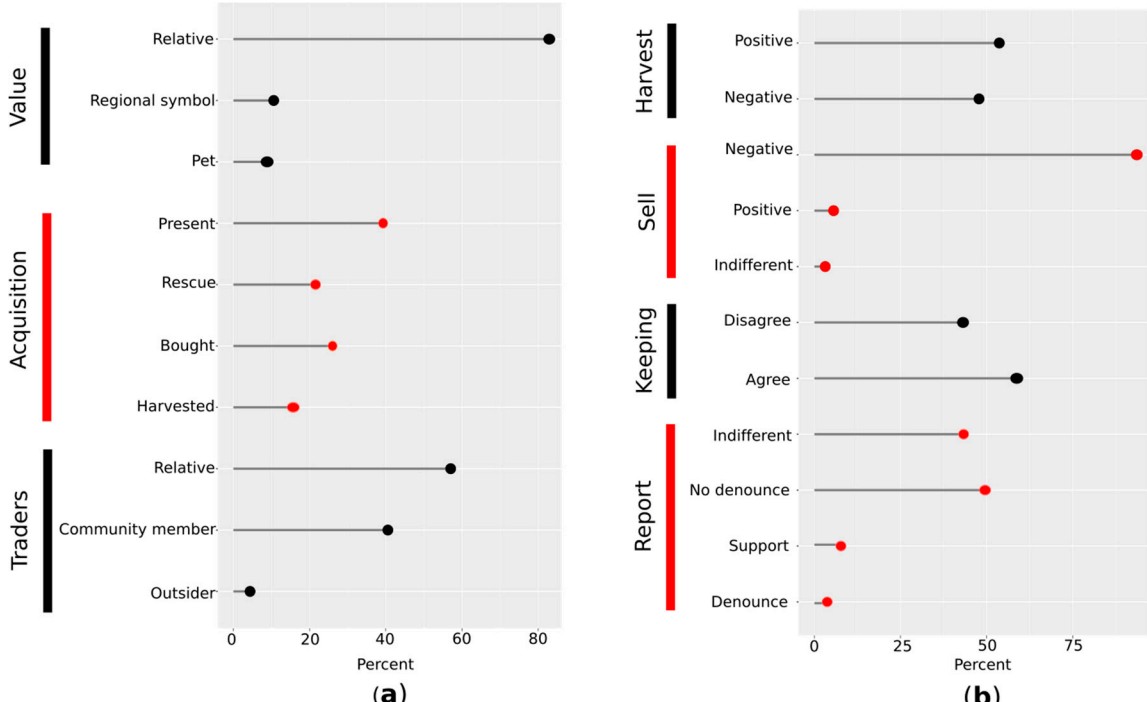

**Figure 4.** Social processes affecting conservation actions implemented by Provita's yellow-shouldered Amazon Conservation Program in Macanao: (**a**) social values and (**b**) attitudes towards stages in the trade chain.

More than half of participants (53%) had positive attitudes towards poaching, as they mostly considered it as a "child prank", an additional source of income, or a tradition (Figure 4b). Participants with negative attitudes towards poaching (47%) considered this practice as "reproachable" or "improper" (Figure 4b).

Almost all participants had negative attitudes towards selling parrots (93%; Figure 4b). Sixty percent of participants showed positive attitudes towards keeping parrots, but a significant proportion (40%) also disagreed with this behavior (Figure 4b). The mean price was USD 1.70 (value range USD 0.30–7.10).

## 4. Discussion

Information on the perceptions and attitudes of local communities is important to identify strategies that best suit conservation objectives, alongside the social and cultural context of local communities that use wildlife. The yellow-shouldered Amazon Conservation Program in Macanao has influenced both positive relations between conservation practitioners and local communities, as well as positive perceptions and attitudes towards species conservation. The absence of a historic baseline prevents before–after or control–treatment comparisons, but in general Macanao communities recognized the leading role of Provita, and perceived that their efforts are articulated by species conservation needs.

Social acceptance of the yellow-shouldered Amazon Conservation Program relates to trust and the long-term commitment of Provita. Although the approval of the program is high, it failed to engage and empower local communities in conservation activities. People were not willing to participate in spite of their high conservation awareness and positive attitudes toward the species.

Future efforts in Macanao will require a stakeholder engagement strategy that fosters high quality management decisions and cost-effective implementations [24]. Given the heterogeneity and changing socio-economic conditions in Macanao, this strategy should include identification of stakeholder relationships and understanding of the incentives to promote their participation. A key stakeholder in

Macanao are the Ecoguardians, a cooperative of local young people that implements conservation actions in the field, mostly related to monitoring and surveillance of parrot nests during the breeding season. The scheme has successfully converted ex-harvesters into parrot protectors [20]. As they are of the age and background of current harvesters, this was perceived as an opportunity for Ecoguardians to become "peer multipliers" of positive conservation attitudes among poachers and the general community [20]. However, half of the community believe them to be poachers. Social network analysis could serve to map the relationships between the Ecoguardians and other stakeholders [25], and these insights, combined with social marketing campaigns, could be used to improve the image of Ecoguardians within the communities of Macanao.

Development of an engagement strategy in Macanao will require identification of the most effective incentive to promote stakeholder participation. Economic benefits are the most frequent incentive [24]. The yellow-shouldered Amazon Conservation Program offered communities alternative livelihoods as Ecoguardians. More than 90 local young people have received income in the last 16 years. Expanding the current strategy to include volunteers, using personal and social benefits as incentives, may enhance parrot breeding success. People can participate in surveillance and enforcement, annual censuses, and building nest boxes to foster parrot nesting opportunities [20].

*Social Processes in the Yellow-Shouldered Amazon Trade*

Evaluation of perceptions also allowed us to interpret how people understand the trade chain and what their role is in it. People from Macanao were aware of the yellow-shouldered Amazon population decline, and they clearly identified unsustainable use as the main cause. However, people seem to relate unsustainable use only to selling, but not to harvesting and keeping.

Selling was the main motivation to harvest fledgling Amazons. Our data partly supports the notion that harvesting for profit is a primary motivation. Although yellow-shouldered Amazons are relatively inexpensive compared to other *Amazona* species in South American markets (e.g., the mealy parrot (*Amazona farinose*) sells for between USD 500–875 in Bolivia) [26], their trade is an opportunity to improve family income through an activity with low risk. However, the profile of poachers does not typically coincide with those of commercial intermediaries [27]. In contrast, in our study poachers kept the parrot as a pet, and professed the same affectionate, non-utilitarian values as keepers. If a sub-group of organized poachers with profit motivation exists, they will require different anti-harvesting strategies than the rest of the poacher community. The fact that people had opposite perceptions about sellers and poachers suggests that different categories (poacher-keepers, poacher-sellers) occur in Macanao, and social acceptability of both actors may differ [28]. For example, "poacher-keepers" could be those harvesting parrots for personal consumption or for relatives and friends. On the other hand, "poacher-sellers" will be able to sell parrots locally, and have the contacts and logistics to sell them in the rest of the country. The former may be more socially accepted than the latter, but they will not report each other because there are relatives involved or fear of retaliation. Future studies should account for the different categories of poachers, their diverse motives, and their role in the illegal trade network, in order to design more effective conservation interventions [9].

As expected, keeping was a widely accepted behavior, but the fact that a significant proportion of interviewees did not agree with parrot keeping suggests that the conservation program has successfully increased conservation concern among people, and there is a good opportunity to change consumptive behavior [29,30]. Local people's knowledge about species conservation status and threats is likely the result of the environmental education program implemented over the last 31 years in 13 schools across Macanao [20,21]. By itself, high levels of knowledge and awareness are not enough for people to engage in actions relevant to species conservation, yet this is the ideal scenario for implementing behavioral change campaigns focused on reducing the unsustainable use of wildlife [31].

Attitudes more consistent with conservation objectives could act as a "seed" in a behavioral change campaign focused on promoting more sustainable uses. For example, citizen science and volunteer programs might help reduce parrot demand while conservation awareness is increased.

"Seed" members may recruit new members into their social networks, who subsequently encourage additional people to participate, and so on [32,33].

The contradiction between improving attitudes and continued high levels of unsustainable parrot harvesting suggests that law enforcement should remain a central activity of the species conservation program. However, the program must not over-rely on enforcement measures, because they fail to address social and cultural factors driving the parrot trade. We suggest that a more bottom-up approach that recognizes the views and motivations of local actors and promotes their engagement in conservation actions could translate to not only lower harvesting and keeping rates, but also in an increase of public support and successful implementation.

## 5. Conclusions

This study contributes to parrot conservation by incorporating perceptions and attitudes to improve adaptive and evidence-based conservation programs. By improving our understanding of key systemic drivers, we are now able to design demand-focused interventions to better tackle the illegal trade of the yellow-shouldered Amazon in Macanao. The fact that despite high conservation awareness and understanding of threats to parrots, people still fail to link their consumption with the illegal trade chain, suggests that a greater effort is needed to demonstrate behavior's impact in conservation, and importantly, which other behavioral options people can pursue to reduce that impact [34]. Behavioral change campaigns based on social marketing must be evidence-based [35], but also nuanced, with cultural, social, logistic, and funding challenges in order to set realistic objectives and efficient outcomes [36]. In the particular case of the yellow-shouldered Amazon trade in Macanao, such nuanced views will imply (1) improved stakeholder engagement strategies, to both manage conflicts and incentive participation and empowerment; (2) the creation of flexible and creative implementation strategies that take into account the widespread poverty, as well as the prevalence of adult women, who are mostly single mothers; and (3) future research to improve our understanding of different categories of harvesters, as well as their motives and role in the illegal trade network [28].

**Author Contributions:** Conceptualization, A.S.-M., O.B., B.S.-S., and J.P.R.; data curation, O.B.; formal analysis, A.S.-M.; investigation, O.B., J.M.B.-L., and C.P.; methodology, A.S.-M.; writing–original draft, A.S.-M. and O.B.; writing–review & editing, A.S.-M., B.S.-S., J.M.B.-L., C.P., and J.P.R. All authors have read and agreed to the published version of the manuscript.

**Funding:** This research was funded by the Whitley-Segré Conservation Fund, the Kilverstone Wildlife Charitable Trust, and the World Land Trust.

**Acknowledgments:** We are grateful to the anonymous participants in this research for their cooperation, and to Hato San Francisco–Arenera La Chica, Fogones y Banderas, the Macanao Marine Museum, and Fundefir-Bankomunal for their support during field work.

**Conflicts of Interest:** The authors declare no conflict of interest.

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
