# Peer review of "Using Peoples’ Perceptions to Improve Conservation Programs: The Yellow-Shouldered Amazon in Venezuela"

_diversity, doi:10.3390/d12090342_

Round 1
Reviewer 1 Report
This is a well designed and executed study on a very important problem in conservation.
My only suggestion was that there is more literature in the challenges of applying basic science outside of conservation and i believe the paper could have broader appeal if this literature was cited at least through one or two review references. This is not essential for the paper but would add value.Author Response
Author’s answer: We appreciate reviewer 1’s comments. We are not sure what reviewer 1 means with “challenges of applying basic science outside of conservation”. We understand this refers to challenges of applying social science into conservation research and practice. We have added three references to address this topic (L39; Bennet et al 2017, Schultz 2011; Dobson et al 2019).
Reviewer 2 Report
Overall, I thought this was a thoughtful and interesting manuscript and I enjoyed reading it. I think it would be of interest to the readers of Diversity and parrot conservationists generally. With a few minor adjustments, I believe this manuscript will be ready for publication.
While I enjoyed the writing style of the manuscript, I think the manuscript would benefit from another round of thorough edits with an emphasis on correcting grammar errors and typos. I detail some of the errors below, but there are many more I did not comment on.
I think the only real issue I had with this paper is that the authors did not introduce the conservation agencies and programs very well in the introduction. Even after reading the entire manuscript, I do not have a clear idea who Provita is – are they an NGO? A government agency? Similarly, the Ecogaurdians were not really introduced fully until line [225-226]. I recommend including a more description of these programs or agencies in the introduction, and what role they have in conserving this species.
Here, I comment on specific line numbers:
[47-48] No study has formally evaluated which order of birds is “most at risk” and there are certainly other orders that are at least AS at risk as parrots (e.g. Procellariiformes and Galliformes), so I recommend rephrasing to just point out that 28% of listed as threatened (or worse) according to the IUCN.
And, as a side note, for parrots, IUCN data is largely outdated – see Martin et al. 2014, Berkunsky et al. 2017, Olah et al. 2018, which all rely on more updated data from the IOU.
[51] recommend adding this citation: (Romero-Vital et al. 2020)
[55-56, 115, 209, 216, etc.] change to “Yellow-shouldered Amazon”
[76] change “with” to “in”
[89-96] The authors alternate between active and passive voice in this paragraph but are largely consistent in using active voice in the rest of the methods. I recommend adjusting the passive sentences to active.
[98 + 101, etc.] Authors are inconsistent with their use of the Oxford comma. If there is no recommendation in the journal’s author guidelines, I recommend picking to either use it or not and being consistent.
[127-128] Repeated reference to the percentage of subjects that graduated high school. Recommend adding phrase “while 26% have university studies [sic]” up to the sentence that ends on line [124]. Also recommend change the phrase to read “while 26% went to university.”
[199] US currency is written to two decimal places. Recommend changing to US$ 1.70. Also, please define what the number in parentheses represent (confidence intervals? standard deviations?) and list them to two decimal places.
[247] change “interpreted” to “to interpret”
[255] put scientific name in brackets and change “is sold for” to “sells for between”
[258] change “bird as pet” to “parrot as a pet”
[259] change “non utilitarian” to “non-utilitarian.”
[281] change “consonant” to “consistent.”
References:
Berkunsky, I, P. Quillfeldt, D.J. Brightsmith, M.C. Abbud, J.M.R.E. Aguilar, U. Alemán-Zelaya, R.M. Aramburú, A. Arce Arias, R. Balas McNab, T.J.S. Balsby, J.M. Barredo Barberena, S.R. Beissinger, M. Rosales, K.S. Berg, C.A. Bianchi, E. Blanco, A. Bodrati, C. Bonilla-Ruz, E. Botero-Delgadillo, S.B. Canavelli, R. Caparroz, R.E. Cepeda, O. Chassot, C. Cinta-Magallón, K.L. Cockle, G. Daniele, C.B. de Araujo, A.E. de Barbosa, L.N. de Moura, H. Del Castillo, S. Díaz, J.A. Díaz-Luque, L. Douglas, A. Figueroa Rodríguez, R.A. García-Anleu, J.D. Gilardi, P.G. Grilli, J.C. Guix, M. Hernández, A. Hernández-Muñoz, F. Hiraldo, E. Horstman, R. Ibarra Portillo, J.P. Isacch, J.E. Jiménez, L. Joyner, M. Juarez, F.P. Kacoliris, V.T. Kanaan, L. Klemann-Júnior, S.C. Latta, A.T.K. Lee, A. Lesterhuis, M. Lezama-López, C. Lugarini, G. Marateo, C.B. Marinelli, J. Martínez, M.S. McReynolds, C.R. Mejia Urbina, G. Monge-Arias, T.C. Monterrubio-Rico, A.P. Nunes, FdP Nunes, C. Olaciregui, J. Ortega-Arguelles, E. Pacifico, L. Pagano, N. Politi, G. Ponce-Santizo, H.O. Portillo Reyes, N.P. Prestes, F. Presti, K. Renton, G. Reyes-Macedo, E. Ringler, L. Rivera, A. Rodríguez-Ferraro, A.M. Rojas-Valverde, R.E. Rojas-Llanos, Y.G. Rubio-Rocha, A.B.S. Saidenberg, A. Salinas-Melgoza, V. Sanz, H.M. Schaefer, P. Scherer-Neto, G.H.F. Seixas, P. Serafini, L.F. Silveira, E.A.B. Sipinski, M. Somenzari, D. Susanibar, J.L. Tella, C. Torres-Sovero, C. Trofino-Falasco, R. Vargas-Rodríguez, L.D. Vázquez-Reyes, T.H. White, S. Williams, R. Zarza, J.F. Masello, 2017. Current threats faced by Neotropical parrot populations. Biological Conservation 214:278-287.
Martin RO, Perrin MR, Boyes RS, Abebe YD, Nathaniel D, Asamoah A, Bizimana D, Bobo KS, Bunbury N, Brouwer J, Diop MS, Ewnetu M, Fotso RC, Garteh J, Holbech LH, Madindou IR, Maisels F, Mokoko J, Reuleaux A, Symes C, Tamungang S, Taylor S, Valle S, Waltert M, Wondafrash M. 2014. Research and conservation of the larger parrots of Africa and Madagascar: a review of knowledge gaps and opportunities. Ostrich 85: 205-233.
Olah, George, Jörn Theuerkauf, Andrew Legault, Roman Gula, John Stein, Stuart Butchart, Mark O’Brien & Robert Heinsohn. 2018. Parrots of Oceania – a comparative study of extinction risk, Emu - Austral Ornithology, 118:1, 94-112
Romero-Vidal, P., Hiraldo, F., Rosseto, F., Blanco, G., Carrete, M., Tella, J.L. 2020. Opportunistic or Non-Random Wildlife Crime? Attractiveness rather than Abundance in the Wild Leads to Selective Parrot Poaching. Diversity 12: 314.
Author Response
Overall, I thought this was a thoughtful and interesting manuscript and I enjoyed reading it. I think it would be of interest to the readers of Diversity and parrot conservationists generally. With a few minor adjustments, I believe this manuscript will be ready for publication.
While I enjoyed the writing style of the manuscript, I think the manuscript would benefit from another round of thorough edits with an emphasis on correcting grammar errors and typos. I detail some of the errors below, but there are many more I did not comment on.
I think the only real issue I had with this paper is that the authors did not introduce the conservation agencies and programs very well in the introduction. Even after reading the entire manuscript, I do not have a clear idea who Provita is – are they an NGO? A government agency? Similarly, the Ecogaurdians were not really introduced fully until line [225-226]. I recommend including a more description of these programs or agencies in the introduction, and what role they have in conserving this species.
Author’s answer: We appreciate reviewer 2’s comments. We did our best to correct grammar errors and typos across the manuscript. We agree with reviewer 2 that our description of Provita and their conservation program was weak. We added several sentences in the introduction section, describing Provita and providing more details about their conservation program and Ecoguardians role (L61 - L66).
Here, I comment on specific line numbers:
[47-48] No study has formally evaluated which order of birds is “most at risk” and there are certainly other orders that are at least AS at risk as parrots (e.g. Procellariiformes and Galliformes), so I recommend rephrasing to just point out that 28% of listed as threatened (or worse) according to the IUCN.
And, as a side note, for parrots, IUCN data is largely outdated – see Martin et al. 2014, Berkunsky et al. 2017, Olah et al. 2018, which all rely on more updated data from the IOU.
Author’s answer: We have deleted the sentence “it is the most threatened bird order globally, with.” We have also replaced the reference of IUCN by Berkunsky et al. 2017.
[51] recommend adding this citation: (Romero-Vital et al. 2020)
Author’s answer: Thanks! We have included it (L50).
[55-56, 115, 209, 216, etc.] change to “Yellow-shouldered Amazon”
Author’s answer: We have replaced it by the proper name.
[76] change “with” to “in”
Author’s answer: We have changed this word as requested.
[89-96] The authors alternate between active and passive voice in this paragraph but are largely consistent in using active voice in the rest of the methods. I recommend adjusting the passive sentences to active.
Author’s answer: We have changed to active voice in this paragraph.
[98 + 101, etc.] Authors are inconsistent with their use of the Oxford comma. If there is no recommendation in the journal’s author guidelines, I recommend picking to either use it or not and being consistent.
Author’s answer: We have kept the use of the Oxford comma. We have checked we used it consistently across the manuscript.
[127-128] Repeated reference to the percentage of subjects that graduated high school. Recommend adding phrase “while 26% have university studies [sic]” up to the sentence that ends on line [124]. Also recommend change the phrase to read “while 26% went to university.”
Author’s answer: We have changed the sentence as requested.
[199] US currency is written to two decimal places. Recommend changing to US$ 1.70. Also, please define what the number in parentheses represent (confidence intervals? standard deviations?) and list them to two decimal places.
Author’s answer: We have added the two decimal places and included “value range” to describe what the numbers in parentheses represent.
[247] change “interpreted” to “to interpret”
Author’s answer: We have corrected the word accordingly.
[255] put scientific name in brackets and change “is sold for” to “sells for between”
Author’s answer: We have added the brackets and changed the sentence accordingly.
[258] change “bird as pet” to “parrot as a pet”
Author’s answer: We have changed the words accordingly.
[259] change “non utilitarian” to “non-utilitarian.”
Author’s answer: We have changed the word accordingly.
[281] change “consonant” to “consistent.”
Author’s answer: We have changed the word accordingly.
Reviewer 3 Report
I have attached a commented version of your paper which will give specific feedback. Overall, some improvement in the details provided in the methods section will address many of the concerns raised later in the paper.

Author Response
I have attached a commented version of your paper which will give specific feedback. Overall, some improvement in the details provided in the methods section will address many of the concerns raised later in the paper.
Author’s answer: We appreciate reviewer 3’s comments. We have added several sentences in the methods section, providing more details about the questionnaire (L107 – L151). Please, see our detailed reply bellow, in the comment corresponding to the lines L104 – L112.
L27 – L29: Rephrase or clarify this sentence.
Author’s answer: We have changed this sentence and hope is clearer now: “Harvesters with different motivations (keepers, sellers) may occur in Macanao, and social acceptability of both actors may differ.”
L104 – L112: More detail is required on this aspect of the methods. Much of the interesting content in the results and discussion are dependent on these (presumably) semi-structured interviews but as currently presented it there is insufficient detail to make clear how the data is generated.
Much more clarity in this section would greatly improve the paper.
Author’s answer: We agree with reviewer 3 that this section lacks clarity. We have added several sentences (L107 – L151) describing the type of questions used (open or closed) and how we assessed or reclassified the answers. Some of the measures we used, such as surveillance effectiveness, were assessed through indirect questions in order to increase the chance of getting reliable answers. In those cases, we described the assumptions under which the relationship between question and measure was based.
L117: This is the only mention of sand miners in the paper - remove or provide some context.
Author’s answer: We have removed it.
L142 – L143: How was this determined? The question in the survey only asks if there will be extinction in 10 years which will give a yes/no answer. I assume that there are follow up questions in the interview but this is not clear in the methods.
Author’s answer: Yes, indeed we used a closed question to assess perceived extinction risk “Do you think that the wild parrot population will go extinct in the next 10 years?” We have replaced “in the short term” by “in 10 years” and deleted the sentence “while 47% thought it may occur in the medium-term” to avoid misunderstandings.
L145: How is this determined? Survey only addresses where is the main location for extraction and not why this is the case. Again I assume that there are follow up questions in the interview but this is not clear in the methods.
Author’s answer: To measure surveillance effectiveness we used a closed question “Where do you think your parrot comes from?” With the names of the most important nesting sites as options, including La Chica. We assumed that the lower frequency on which La Chica was mentioned as an extraction site compared to other nesting sites, is a good indirect measure of surveillance effectiveness, because La Chica has been the only nesting site under protection during the last 30 years. We have added some sentences in the methods section to clarify this point (L118 – L121).
L145 – L146: Looks lower in the figure (60%?) Is this a typographical error?
Author’s answer: Yes, this is a typo error, the correct values are 39% in La Chica and 61% in other nesting sites. We have corrected it.
L149: surveillance?
Author’s answer: We have corrected this typo error in the Figure 2.
L156: How is this determined?
Author’s answer: Given that Ecoguardians are a key stakeholder, we inquired about perceptions towards Ecoguardians with an open question “What do you think about the work of Ecoguardians?” and then reclassify the answers into positive and negative perceptions. We have added some sentences in methods section (L129 – L131) to better explain this.
L198 -L199: How was this determined? Needs to be made clear in the methods.
Author’s answer: We asked how much their parrots were worth in national currency and then, we transformed it into US$ using the weekly mean of currency exchange rate. We have added some sentences in methods section (L149 – L151) to better explain this.
L203: Harvest
Author’s answer: We have corrected this typo error in the Figure 4.
L252 – L260: This discussion is not well introduced in either the methods or results. Some more clarity earlier in the paper will make this section more appropriate.
Author’s answer: We have addressed this point in our answer to reviewer 3 general’s comment. We have added several sentences in the methods section to better describe the social processes in the Yellow-shouldered Amazon trade (L136- L149).
L277 – L280: Rephrase/clarify this sentence.
Author’s answer: We have rewritten this sentence and now reads “By itself, high levels of knowledge and awareness are not enough for people to engage in actions relevant to species conservation, yet this is the ideal scenario for implementing behavior change campaigns focused on reducing unsustainable use of wildlife.”